# BERNN: Enhancing classification of Liquid Chromatography Mass Spectrometry data with batch effect removal neural networks

Simon J. Pelletier[1], Mickaël Leclercq[1], Florence Roux-Dalvai [1,2], Matthijs B. de Geus [3,4], Shannon Leslie[5,6], Weiwei Wang[7], TuKiet T. Lam [7,8], Angus C. Nairn [5], Steven E. Arnold[3], Becky C. Carlyle [3,9,10], Frédéric Precioso[11] & Arnaud Droit [1,2] ✉

Liquid Chromatography Mass Spectrometry (LC-MS) is a powerful method for profiling complex biological samples. However, batch effects typically arise from differences in sample processing protocols, experimental conditions, and data acquisition techniques, significantly impacting the interpretability of results. Correcting batch effects is crucial for the reproducibility of omics research, but current methods are not optimal for the removal of batch effects without compressing the genuine biological variation under study. We propose a suite of Batch Effect Removal Neural Networks (BERNN) to remove batch effects in large LC-MS experiments, with the goal of maximizing sample classification performance between conditions. More importantly, these models must efficiently generalize in batches not seen during training. A comparison of batch effect correction methods across five diverse datasets demonstrated that BERNN models consistently showed the strongest sample classification performance. However, the model producing the greatest classification improvements did not always perform best in terms of batch effect removal. Finally, we show that the overcorrection of batch effects resulted in the loss of some essential biological variability. These findings highlight the importance of balancing batch effect removal while preserving valuable biological diversity in large-scale LC-MS experiments.

Liquid chromatography-mass spectrometry (LC-MS) has become an essential analytical technique in molecular biology because of its ability to accurately and simultaneously quantify thousands of compounds in biological samples. It is a powerful tool for large-scale screening of potential biomarkers, which enables the identification of specific measurable indicators that can aid in disease diagnosis[1], prognosis[2], and treatment selection[3], leading to improved patient outcomes and personalized medicine. Despite the power of LC-MS, the utility of large-scale experiments remains compromised due to the omnipresence of confounding factors. Confounders can be divided into biological confounders, such as age or gender, and non-biological confounders, such as batch effects. The latter are practically

[1]Computational Biology Laboratory, CHU de Québec - Université Laval Research Center, Québec City, QC, Canada. [2]Proteomics Platform, CHU de Québec - Université Laval Research Center, Québec City, QC, Canada. [3]Massachusetts General Hospital Department of Neurology, Charlestown, MA, USA. [4]Leiden University Medical Center, Leiden, The Netherlands. [5]Yale Department of Psychiatry, New Haven, CT, USA. [6]Janssen Pharmaceuticals, San Diego, CA, USA. [7]Keck MS & Proteomics Resource, Yale School of Medicine, New Haven, CT, USA. [8]Yale School of Medicine, Department of Molecular Biophysics and Biochemistry, New Haven, CT, USA. [9]Oxford University Department of Physiology Anatomy and Genetics, Oxford, UK. [10]Kavli Institute for Nanoscience Discovery, Oxford, UK. [11]Université Côte d'Azur, CNRS, INRIA, I3S, Sophia Antipolis, Nice, France. ✉e-mail: arnaud.droit@crchudequebec.ulaval.ca

unavoidable in large-scale studies due to limitations in instrumental availability and timeline of sample collection and ideally would be removed from the final biological quantification value[4]. It can be difficult, even impossible, to completely remove batch effects without affecting the quality of the biological signal. By assessing the classification improvement of machine learning models, we can determine if the batch effect correction method successfully removes the technical variations and restores the underlying biological patterns[5,6]. Furthermore, classification models enable personalized medicine by identifying patterns and biomarkers that can discriminate between different subjects. In highly heterogeneous disorders, such as Alzheimer's Disease, it enables the discovery of biomarkers that are not common to every person affected by this disorder[7]. Batch effects are a systematic variation that arises from experimental differences introduced unintentionally during data collection. It typically emerges from differences in sample processing protocols (e.g., variations in technicians, reagents, or equipment), experimental conditions (e.g., discrepancies in temperature, and humidity), and data acquisition techniques (e.g., variations in sequencing platforms, microarray scanners). It is a critical problem in many high-throughput bioassays, such as microarrays[8], RNA-Seq[9] or LC-MS-based proteomics and metabolomics[10], that involve processing samples in different batches or on different platforms, which can result in differences in data quality and analysis performance. Batch effects in proteomics occur due to variability in sample preparation, instrument condition, and performance, or environmental factors present during the sample preparation and analysis workflow. These technical variations may lead to false positive or false negative protein identifications, as well as inconsistencies in the quantification of protein abundances across batches, which can hamper the reproducibility and validity of the study[11].

Common approaches to correct batch effect in LC-MS include Quality Control (QC) based methods (e.g., qc-rlsc[12]), location-based methods (e.g., Combat[8]), and matrix refactorization methods (e.g., harmony[13]). These methods assume an accurate new representation can be obtained using a generalized linear model, which is not necessarily an accurate assumption because they have a limited capacity and might not capture the whole complexity of the batch effect[14]. This issue can be addressed using Deep Neural Networks (DNN), which use a succession of nonlinear transformations that enable them to correct more complex batch effects[14]. In the latter, batch information is incorporated into the DNN either as an input feature[6,14,15], as a regularization term in the objective function[16], or vectorized and added to the encoded vector representing the input features, such as NormAE[12–14]. However, because of their over-parameterization, DNNs usually require large amounts of data to be generalizable. Given the high number of parameters to train and the fact that there is a high number of possible architectures (and so no one-fits-all solution), DNNs can be resource-intensive and time-consuming to train. To counter this problem, we used careful model selection and regularization methods (we used weight decay, dropout, and label smoothing). We used a repeated holdout scheme where we resampled the validation and test set up to 5 times to reduce the possibility of reducing the chance of getting good results by chance.

To determine whether a biological signal has been preserved after batch effect correction, one can utilize classification performance. The highest scores determine the effectiveness of the correction. There are several metrics available to evaluate the extent to which the batch effect has been removed, with the goal being to determine an optimal balance between preserving biological variation and removing the batch effect[6]. The success of batch effect removal is often evaluated based on the number of significant features using differential analysis methods[11,17] or solely on the reduction of batch-mixing metrics[10,18]. Discrimination between classes under study using DNNs might be discredited because they are considered black-boxes[19] and the purpose is usually to find biomarkers for disorders that can already be diagnosed otherwise. Today, the problem of interpretability of DNNs has been thoroughly investigated and many options are available to break into the black box[20,21], including methods specific to deep learning models, such as attribution methods (e.g., Integrated Gradients[22] and DeepLIFT[23]), model-agnostic approaches (such as LIME[24] and SHAP[25]), all of which provide insight into how each feature is important for the model's decision-making process. Not only do these methods can indicate the average importance of each feature for a complete classification task, but they can also provide interpretable classification explanations at the sample level[26].

In this context, we propose a suite of models to overcome these various batch correction problems, all based on autoencoders executable in parallel. Our approach to countering batch effects is different from most other solutions, as we do not rely on a single solution that we claim to be superior to all others. Instead, we acknowledge that not all problems require the same solution and propose multiple potential solutions to address batch effects. Thus, we aim to empower researchers to easily try multiple methods simultaneously and pick the optimal approach for their dataset and scientific questions. Amongst this suite of models, we present the first use of Variational Autoencoders (VAE), Domain Adversarial Neural Networks (DANN), and Domain Inverse Triplet Loss (invTriplet) for batch correction in LC-MS. Additionally, in contrast to other batch correction methods, we do not recommend using the corrected output of the autoencoder for biomarker discovery through downstream analysis (e.g. using differential analysis). Instead, we demonstrate how SHAP[25] can be used for biomarker discovery. Finally, our method simultaneously corrects batch effects and performs sample classification, making the method an end-to-end solution that ensures that batch effect removal improves classification.

## Results
### Model descriptions
Our models, which we call Batch Effect Removal Neural Networks (BERNN), are composed of different modules, each with different objectives: the autoencoder, the batch classifier, and the label classifier (Fig. 1; for more details Supplementary Fig. 1). The autoencoder aims at finding representations, usually smaller than the inputs, that can be used to reconstruct the inputs, as can be seen in Fig. 1A and B. The autoencoder, by itself, can improve classification generalization by removing the noise and by providing a smaller representation. To further remove batch effects, we use either an adversarial loss or a modified version of the triplet loss to find feature representations that cannot discriminate between batches. The adversarial strategy is already used by NormAE[17], but we are using the Gradient Reversal Layer (GRL) to make the training more straightforward[27]. To our knowledge, all methods in the literature using DNN to deal with batch effects use an autoencoder[5,14,15,17], but it is not mandatory (as in ref. [27] original model). The label classifier is optional to get a representation free of batch effect, but is essential for model selection, as we define the objective of the model as to get the best classification scores in batches never seen during training. This objective ensures the maximum biological information is preserved and should therefore always be used to improve the reliability of downstream analysis. To ensure generalizability across batches, we always validate our results on batches that were not used during training. The label classifier was not part of the original NormAE model, which gave it an unfair disadvantage compared to the other models developed in this study. Hence, we modified NormAE to use the same architecture and training scenarios as our models. Our version was superior to the original NormAE (e.g. on the Alzheimer dataset, a 5-fold cross-validation with original NormAE had an average of 0.1 MCC, compared to 0.37 MCC for our version on validation sets).

In addition to the methods used to obtain batch-free representations, we also used Variational Autoencoders (VAE) (see Methods). With recent technologies, datasets tend to have a very high

number of features, but a much smaller number of samples. VAE enables the model to perform data augmentation in datasets that are highly dimensional, but with few samples. VAEs can reduce overfitting in overparameterized neural networks[28], which is important for optimal performance.

Most batch effect correction methods return the corrected expression data, but ref. 29 does not provide the corrected features: it creates an integrated embedding. All our methods also create a new embedded representation but can also return a vector of the corrected data, although the utility of these data is not guaranteed. The corrected data is obtained by reconstructing the original input from the new embedding using the decoder (the purple module in Fig. 1). This is because the importance of the reconstruction loss (i.e., a measure of the error between the predicted and true values) is only one of the

multiple losses that constitute the final loss. Each individual loss can have its importance significantly reduced by a hyperparameter. When used in combination with a classification task, it is possible to get the features that had the most importance for the classification. Although BERNN is designed to optimize classification, it can also be used for biomarker discovery by using SHAP[25], by quantifying the contribution of each feature to the model predictions and thus identifying the most influential features as biomarkers. BERNN provides insights into the relative importance of different features, enabling the identification of biomarkers for improved understanding and diagnosis in various biomedical applications. We ran a SHAP analysis (See Supplementary Fig. 2) as an example of how the features contributing the most to the decision of the models can be identified without the corrected original features.

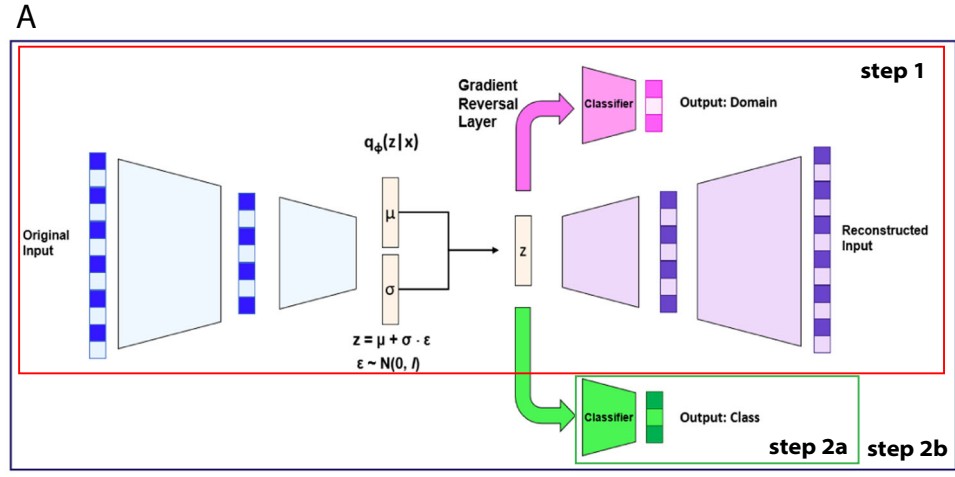

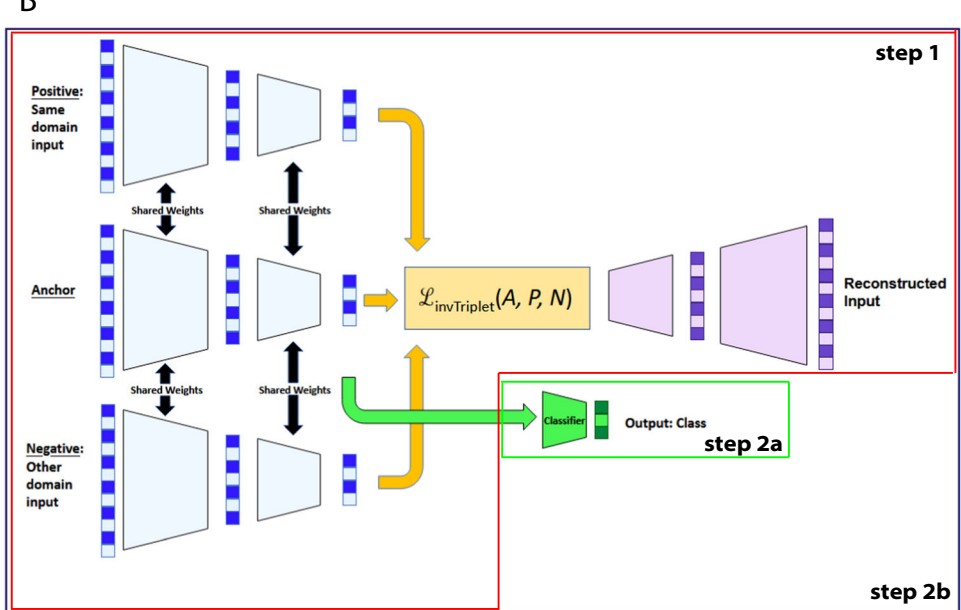

**Fig. 1 | Models architectures. A** VAE-DANN is a variational autoencoder with a domain classifier trained with a gradient reversal layer (GRL), where φ represents the parameters of the encoder (q), the parameters learned by the encoder are μ, σ, and ε, which correspond to the mean, variance and gaussian noise, respectively. **B** AE-invTriplet is an Autoencoder that uses the inverse triplet loss to make the new representation batch-free. The encoders for the positive, negative, and anchor samples all share the same weights. In both models, step 1 is the warmup and uses the whole dataset (including the train, validation, and test sets). Once the warmup is done, two scenarios are possible. In the first possibility, step 2a, only the labels classifier is trained using the training set; the rest of the model is frozen, not allowing backpropagation to go through the encoder. In the second scenario, step 2b, the training alternates between step 1 and step 2a, with the exception that backpropagation is allowed to flow through the encoder. The two models share some of the same modules: the encoder (blue), the decoder (purple), and the labels classifier (green). Some of the modules are not shared between the two models represented here but could be used to make new models. It is the case for the domain classifier (pink), the inverse triplet loss (yellow), and sampling from the parameters of a Gaussian distribution (beige).

## Evaluation and selection

To get an accurate picture of the model performances benchmarked in this study, we used five datasets with different characteristics (Fig. 2). They are different in terms of a number of features (889, 1235, 6461, 17,887, and 18,114), a number of batches (3, 7, 7, 21 and 78) and the degree of batch effect (Adjusted Mutual Information (AMI) from 0.13 to 1.0). The differences in initial batch effects are also visually evident (Fig. 2B). These datasets, which have very different characteristics, achieved top performances with different models, supporting the concept that different problems might require different solutions. Models were evaluated on their classification performances using the accuracy (Supplementary Fig. 3) and Matthews Correlation Coefficient (MCC) (Figs. 3A, 4A, 5A, 6A, Supplementary Fig. 8A). Because some datasets were highly imbalanced (more samples in one class than the other), we selected the top performing models based on their MCC scores. The MCC produces high scores only if predictions are good across all four confusion matrix categories (true positives, false negatives, true negatives, and false positives)[30]. Class imbalance was strongest in the Adenocarcinoma dataset, where 87.5% of the samples in the dataset were of the dominant class. As can be seen in Supplementary Fig. 3B, the best accuracy for this dataset is obtained using the raw data, because the model always predicts the dominant class. The main drawback of using MCC is its high sensitivity to misclassification of the samples in the minority class. When the imbalance is very high, misclassification of a single sample can have a big impact on the score. This can partly explain why the error bars for MCC were sometimes quite large. In comparison, error bars are smaller for accuracies (Supplementary Fig. 3).

To evaluate the performance of batch effect correction, we used two main categories of metrics: batch mixing metrics and quality control (QC) metrics. To assess the batch mixing performance, we used Batch Entropy (BE), Adjusted Rand Index (ARI), and AMI (see Methods). QC metrics measure how much the QC samples are different from each other. Because QC samples are the same for a given dataset, they should in theory be at the same position in an Euclidean space, and each of the features should have perfect Pearson Correlation Coefficients (PCC). The metrics nMED and aPCC are derived from these assumptions, respectively (see Methods). Note that in results presented in Figs. 3–6 and Supplementary Figs. 4–8, for BERNN models, all the batch effect correction metrics are for models where the MCC was maximized. They could all possibly reach much better batch effect correction performance, but it is not the objective of the study.

## Training scenarios

As explained in Fig. 1, there are 2 scenarios to train any BERNN. Both scenarios start with a warming-up phase (step 1), which uses the complete dataset (including the validation and test set) to train all components of the model that are unsupervised, which include the autoencoder and the batch classifier (if an adversarial loss is used). In the first scenario, when the warmup is over, the autoencoder is frozen and only the labels classifier is updated using the training set (step 2a). In the second scenario, after the warmup, the training alternates between step 1 and step 2b, the same as step 2a, except that the backpropagation is allowed to flow through the encoder.

The first scenario was used only for the Alzheimer dataset, the four others used the second scenario. When the second scenario is used with the Alzheimer dataset, the overfitting problem is too important, and only random predictions are made for the validation and test sets. However, because the label classifier cannot influence the representation learned, it causes an underfitting of the data. It may be because of the data augmentation they provide that VAEs, particularly VAE-DANN, perform better on this dataset (Fig. 3A). On the Adenocarcinoma, Aging Mice, Benchmark, and mixed tissues datasets, the second scenario performed best. The first scenario made the model

underfitting too much, thus the second scenario performed much better (Fig. 4A, 5A, 6A, and Supplementary Fig. 8A). In the Adenocarcinoma dataset, the AE-based models performed much better than the VAE-based models, especially AE-invTriplet. For the Benchmark dataset, all models improved the classification performances, with AE-invTriplet having the best classification score (Fig. 6A). For the Aging Mice and Mixed tissues datasets, all BERNN models were better than the other methods, but their performances were almost the same, with only negligible differences (Fig. 5A and S8A). In these cases, it is possible that the maximum performance was obtained even with the simplest AEs, leaving no possibility for improvement with more advanced methods. Thus, we found that no single model can pretend performing optimally for all five datasets analyzed, with some models being the best for a certain dataset while performing poorly on others.

## Reducing batch effects can improve classification

We define the best model as the one with the best average classification performances over all repeated holdout iterations (see Methods). Because we are interested in studying if the model learned generalizes to new batches, all the samples from a given batch must be contained within the same split. In standard cross-validation practices, the test set always remains the same. However, some batches are easier to classify in any given training set, which could explain the sometimes-large error bars in Fig. 3A, 4A, 5A, 6A, and Supplementary Fig. 8A. When using cross-validation, if the test set randomly comprises these "easy" classification batches, it may lead to much better performance in the test set than in the validation set. To avoid this potential confusion, we used a repeated holdout method so that the test set is resampled for each holdout iteration.

Most batch correction methods improve (decrease) normalized batch entropy (nBE). AMI and ARI are also consistently improved (decreased in value) in datasets with the greatest initial batch effects, but not in the Alzheimer dataset, which had moderate batch effects in the raw dataset. For every dataset where the classification performances were not already almost perfect on the raw data, the best validation MCC score always used a transformation that improved nBE (Figs. 3D, 4D, 6C). Although in the case of the adenocarcinoma dataset, many VAE-based networks performed worse than a classification using the raw data (see Methods), VAE and AE networks are both the most efficient in at least one dataset. In all datasets, all the best classification performances were reached using neural networks models that also significantly improved nBE, unless the MCC scores were already near perfect without removing any batch effect. It is also true that some of the methods that led to some of the best nBE improvements had some of the worst MCC scores, sometimes far worse than when using the raw data. The best examples of this are the Aging Mice and mixed tissues datasets, where the classification scores using the raw data reached high MCC scores, despite having high nBE (Fig. 5 & Supplementary Fig. 8). On the other hand, Combat and harmony reduced batch effects at the cost of very low MCC scores. This reduction can lead to the loss of important biological information, which could have negative consequences even in situations where classification is not a primary concern, such as differential quantification/expression analysis. When evaluating differences between samples from two conditions, classification performance provides evidence of the significance of a set of biomarkers identified to discriminate between the conditions.

In both datasets that used replicate QC samples (Alzheimer and Adenocarcinoma), the networks with the best classification performances had better nMED than the uncorrected data, but the networks that did best on these metrics did not necessarily result in the best classification metrics. Note that for all of BERNN models, only the model with the best classification performances were retained. Many models that performed even better on every batch effect metrics than the models kept were discarded due to poor classification performances.

A

| Dataset | Organism | Sample source | Data Type | Sample-to-sample heterogeneity | AMI | Number of batches | Number of samples (excluding pools) | Number of pool samples | Biological factors | # Features | Dataset accession |
|---|---|---|---|---|---|---|---|---|---|---|---|
| Alzheimer | Human | CSF | Proteins | Very high: samples from patients with neurological disorders. Alzheimer's disease is a heterogeneous disorder and the controls have a wide variety of of neurological disorders with no cognitive impairment | 0.13 | 21 | 408 (839 with duplicates) | 84 | Disorder Sex Age | 889 | https://github.com/spell00/BrainBatchEffectMSMS |
| Aging Mice | Mouse | Liver tissue | Peptides precursors | Medium: samples come from population of inbred mice originating from two parental strains | 0.82 | 7 | 372 | 3 (discarded) | Strain Diet Age | 17887 | PRIDE PXD009160 |
| Adenocarcinoma | Human | Plasma | Metabolites | High: samples come from cancer patients | 1 | 3 | 642 | 74 | Disease | 6461 | https://github.com/dengkuistat/WaveICA |
| Mixed tissues | Human & yeast | Human: ovary & prostate cancer tissue, HEK293T cell line. Yeast: BY4741 strain | Peptides | Very low: All samples are present in duplicates or triplicates. All samples are a mix of the same 3 tissue samples in different concentrations (except for the HEK293T control cell line, which is not mixed) | 0.82 | 78 | 8 (1553 with replicates) | 0 | Proportions of mixed tissues | 18114 | PRIDE PXD015912 |
| Benchmark | Bio-molecules and E. Coli | Not Applicable | Metabolites | Very low: high number of replicates. All samples from the same class are the same but at different concentrations | 0.95 | 7 | 72 (1027 with replicates) | 0 | Biomolecules Organism | 1235 | https://www.research-collection.ethz.ch/handle/20.500.11850/545373 |

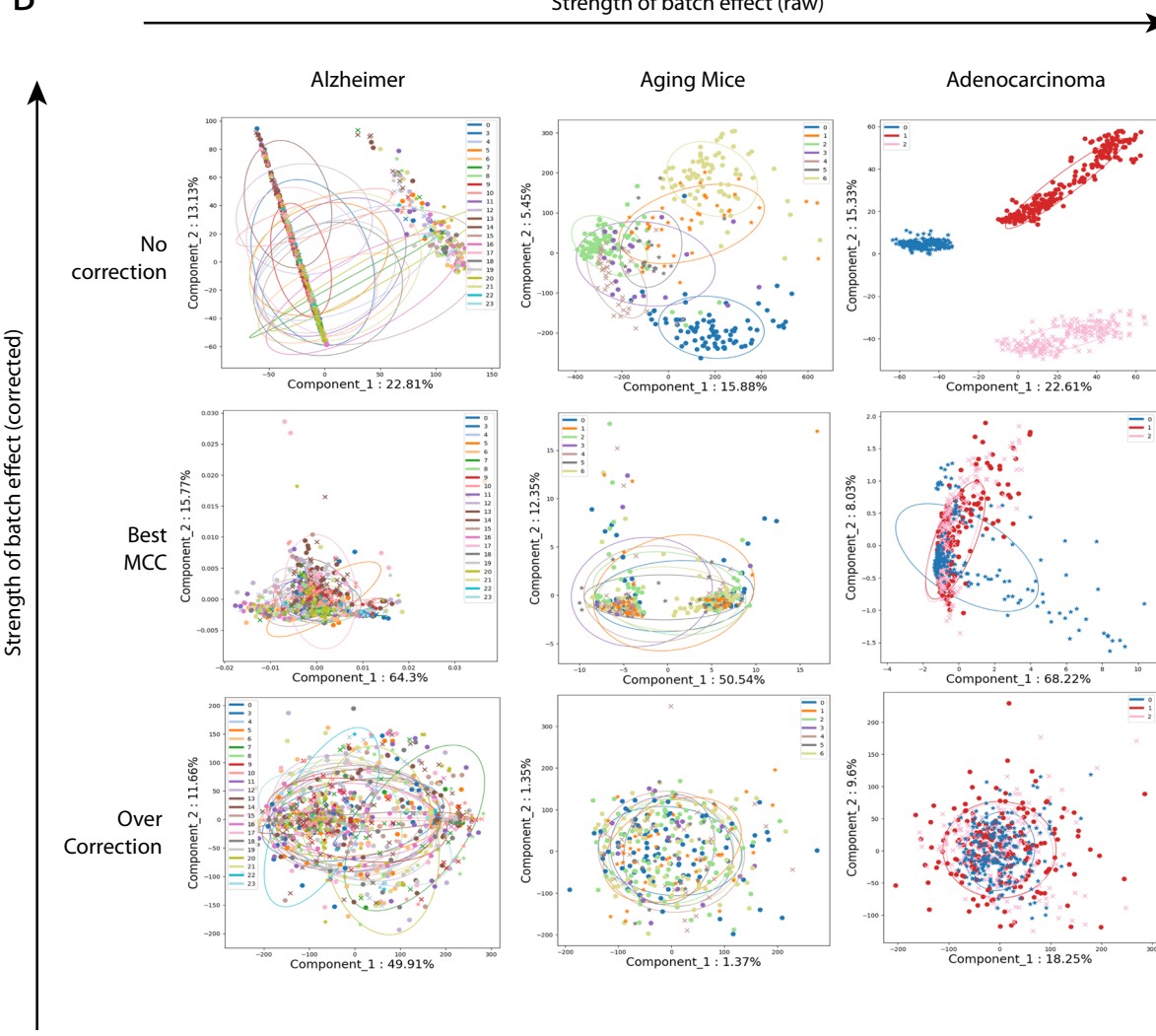

**Fig. 2 | Datasets description and overview of batch effect correction. A** Table summarizing the five datasets used in the study. For each dataset, 2 conditions are classified by BERNNs. **B** PCA visualization of the raw data for five datasets. The datasets are, from left to right, ordered by the strength of the initial batch effect. For each dataset, the middle row of images represents the transformation resulting in the best valid MCC. The images in the last row are from representations that were in the top methods purely for batch correction but performed badly for classification. The PCA visualization of the two datasets not represented here is available in Supplementary Fig. 13.

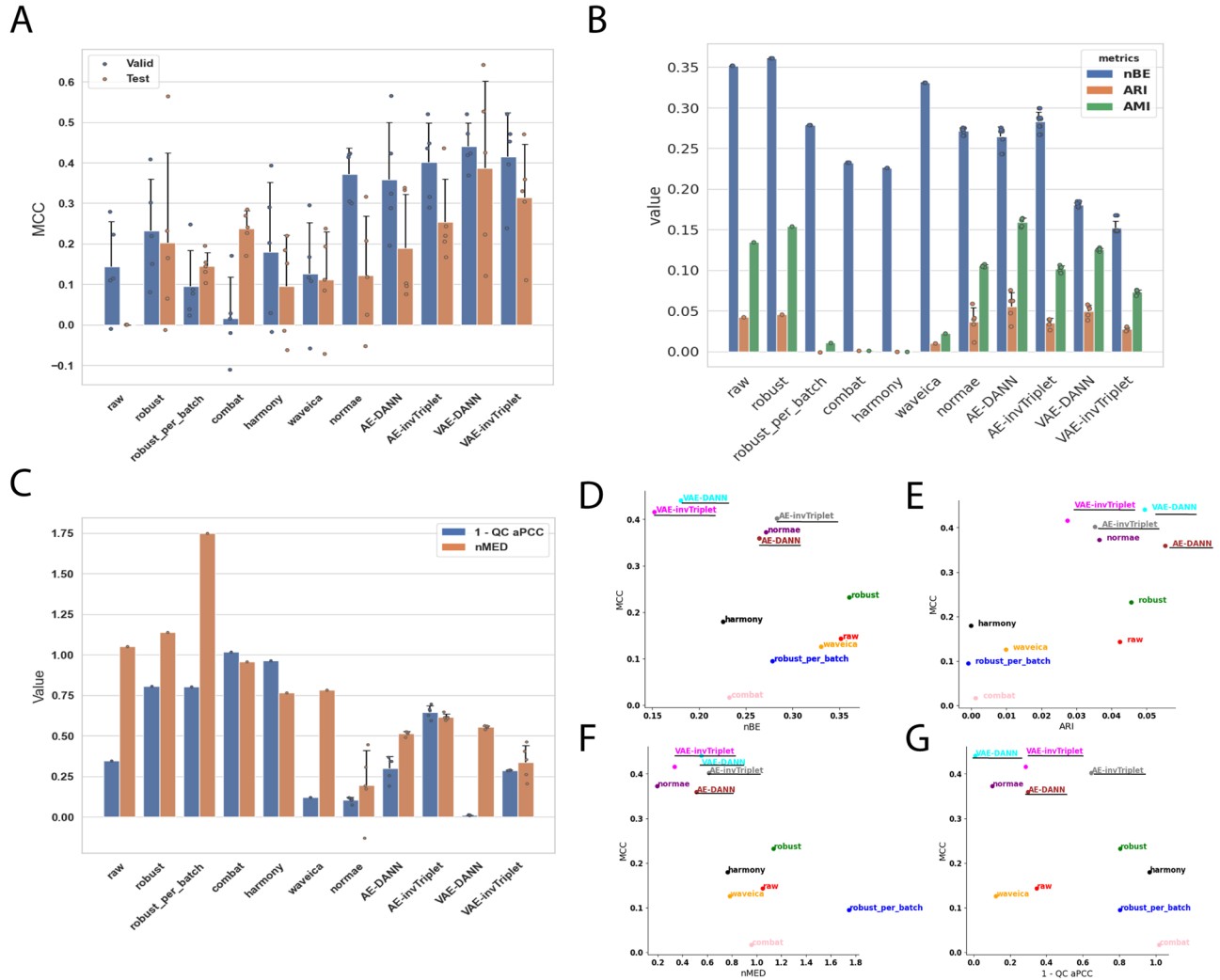

**Fig. 3 | Metrics on the Alzheimer dataset. A** Valid and test MCC scores for all methods benchmarked. Higher is better. The conditions compared are Cognitively unimpaired patients and Alzheimer's Disease with dementia. **B** Batch mixing metrics: normalized Batch Entropy (nBE), Adjusted Rand Index (ARI), Adjusted Mutual Information (AMI). Smaller is better. **C** QC metrics: Normalized Median Euclidean distance (nMED) and QC average Pearson Correlation Coefficient (qc_aPCC). Lower nMED and 1-qc_aPCC are better. MCC is compared to **D** nBE, **E** ARI, **F** QC nMED, and **G** QC aPCC. Error bars represent standard deviations around the means. All error bars are derived from the results of 5-fold cross-validation (*n* = 5). The BERNN models are underlined. Source data are provided as a Source Data file.

## AE outperforms all other methods

While it is not possible to confirm a single model as the best choice to improve classification, all the best MCCs were obtained using a version of our BERNNs. All of them performed almost identically on the Aging Mice and Mixed tissues dataset. On the Alzheimer dataset, VAE-DANN performed the best, followed closely by NormVAE and VAE-invTriplet. It is possible that the reverse triplet loss (revTriplet; for definition see supplementary methods), which did not perform as well as the other models developed in this study, would perform best in other datasets (for the complete results, see Supplementary Figs. 4–9). In all datasets, one of revTriplet or invTriplet was almost always part of the best of the BERNN models for batch correction according to the batch entropy (Supplementary Table 1–5), the only exception being for the Mixed tissues dataset where normvae and VAE-DANN had the lowest nBE. Combat or harmony was often the best according to the three batch mixing metrics, but always at the cost of poor classification performances. The triplet losses require an additional hyperparameter (called *margin*), which makes hyperparameter optimization more complex. It has been shown to have important consequences on optimization in scRNAseq where inverse triplet loss was also used to overcome batch effects[29]. It is possible they might require more hyperparameter optimization to achieve even better performances. Nevertheless, one of the invTriplet models was either the top performer or was always close to the best performance (Supplementary Table 1–5).

When batch effects hindered classification generalization, using one of BERNN's models was always beneficial in all three datasets. However, when a batch effect is evident, but the model can generalize predictions in new batches, the course of action is less clear. The AgingMice dataset provides a good example of this scenario, as LinearSVC achieved particularly good classification scores with unnormalized data. Normalization methods that reduce batch effects or batch correction methods applied prior to training the model classifications often had some of the best batch mixing scores (Supplementary Fig. 8). However, they either reduced the efficiency of the classification tasks or barely made any difference in performance. The classification was significantly reduced by some methods, such as harmony or combat (excluding pycombat), indicating that the biological signal crucial for distinguishing between conditions was completely eliminated.

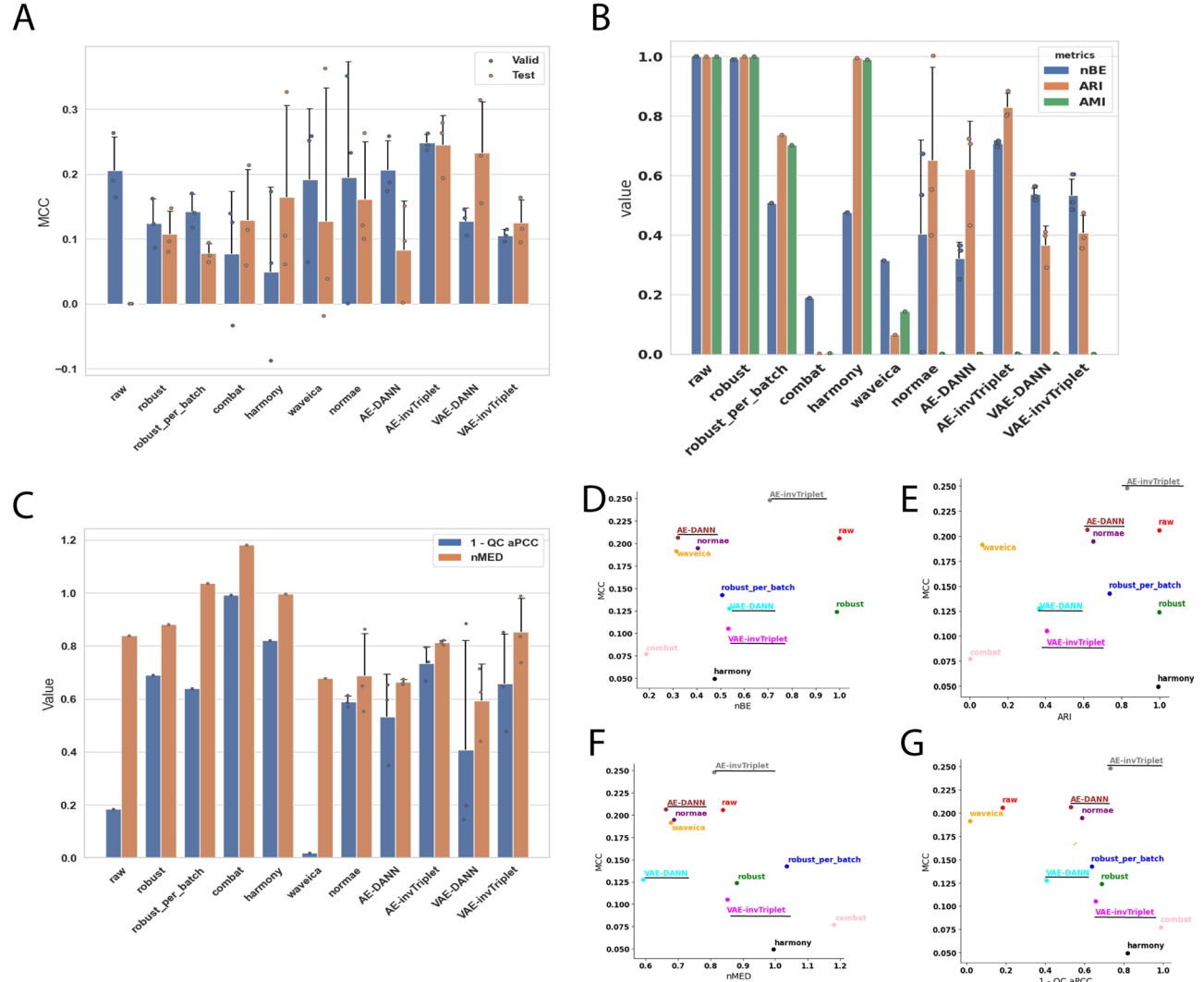

**Fig. 4 | Metrics on the Adenocarcinoma dataset. A** Valid and test MCC scores for all methods benchmarked. Higher is better. The conditions compared are colorectal cancer and chronic enteritis. **B** Batch mixing metrics: normalized Batch Entropy (nBE), Adjusted Rand Index (ARI), Adjusted Mutual Information (AMI). Smaller is better. **C** QC metrics: Normalized Median Euclidean distance (QC nMED) and QC average Pearson Correlation Coefficient (qc_aPCC). Lower nMED and 1-qc_aPCC is better. MCC is compared to **D** nBE, **E** ARI, **F** QC nMED and **G** QC aPCC. Error bars represent standard deviations around the means. All error bars are derived from the results of 3-fold cross-validation ($n = 3$). The BERNN models are underlined. Source data are provided as a Source Data file.

## Multiclass classification

To test BERNN on a multiclass problem, we used two datasets. The first is provided by ref. 31 (description in Methods), which has 6 classes. As seen in Fig. 6, all metrics indicate the presence of a strong batch effect. Similarly to the Alzheimer's and Adenocarcinoma datasets, the model with the highest MCC (in this case AE-invTriplet) decreases batch effects metrics (Fig. 6B–D) compared to classification on the unnormalized data, without being the model that lowers batch effects the most.

The second is provided by refs. 32–35 (description in supplementary Methods), which has 8 classes. As seen in supplementary Fig. 8, all metrics indicate the presence of a moderate batch effect (higher than the Alzheimer dataset, but lower than the Adenocarcinoma or AgingMice datasets) (Supplementary Fig. 8B). As with the AgingMice dataset, the classification of the raw data was nearly perfect even without batch effect correction. All BERNN's models reduced batch effects according to every metric (nBE, ARI, and AMI), but not as much as combat. However, contrary to other batch effect correction methods (combat, harmony, and waveICA), correcting the batch effect with all BERNN's models did not reduce the classification performance (Supplementary Fig. 8A). Though combat is the top performer based

on the batch effect metrics, it is the worst based on the classification performance.

## Discussion

Our contribution to researchers who are facing batch effect problems is threefold. First, we demonstrated the effectiveness of models that, to our knowledge, have never been applied in LC-MS experiments to correct batch effects. Secondly, we showed the necessity of trying different models to solve different problems. Finally, we show that to obtain the best classification on a given dataset, removing parts of the batch effects can improve the results, but removing too many batch effects might come at the cost of diminished classification performance. We argue that the classification of the conditions of interest should be the main objective instead of just removing batch effects. Failure to do so may lead to the loss of biological information, so even if classification is not the main objective of a study impacted by batch effect, we believe that over-correcting batch effects would then negatively impact any downstream analysis, such as differential analysis. Correcting for batch effect should thus be treated as a secondary objective that helps improve a classification task.

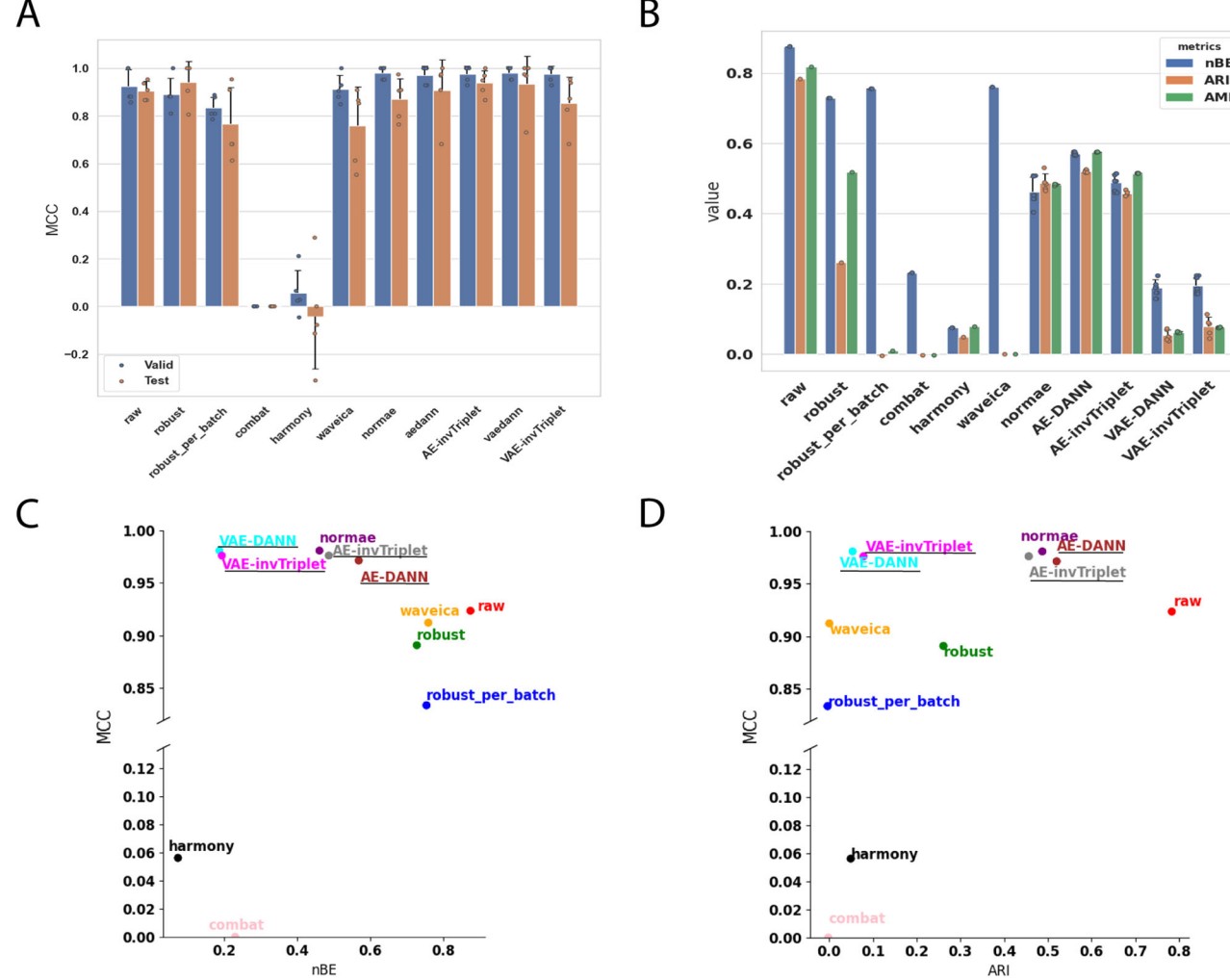

**Fig. 5 | Metrics on the AgingMice dataset. A** Valid and test MCC scores for all methods benchmarked. Higher is better. The conditions compared are mice with high-fat diet and chow diet. **B** Batch mixing metrics: normalized Batch Entropy (nBE), Adjusted Rand Index (ARI), Adjusted Mutual Information (AMI). Smaller is better. MCC is compared to **C** nBE and **D** ARI. Error bars represent standard deviations around the means. All error bars are derived from the results of 5-fold cross-validation (*n* = 5). The BERNN models are underlined. Source data are provided as a Source Data file.

It is well known in machine learning that no single model can claim to be the best at solving any task. In this study, we propose a suite of deep learning architecture models to enable users to find the optimal solution for different problems. This suite introduces batch correction in LC-MS using VAE-based models, the use of GRL for implementing a DANN and triplet losses, all of which were part of the best model in at least one dataset.

The inverse triplet loss is particularly interesting because it is the only loss that is effectively minimized. The other losses that use GRL attempt to minimize the batch classification, but the loss increases until it reaches the loss corresponding to a random classification of the batches. This property is particularly useful in the context of multitask learning[33], because it requires all losses to be minimized to function properly. In this study, some models require to simultaneously train multiple losses: the autoencoder reconstruction loss (1), the batch classification loss (2), the classification loss (3), and, for VAEs, the Kullback-Liebler loss (4). Each of these losses needs hyperparameters to leverage their importance for the model to be optimal. Adversarial models, for example, are notoriously known to be difficult to train because of the fragile balance between the training of the discriminator and generator[34].

The bottleneck representation should be preferred to the reconstructed inputs because the decoder cannot improve the

representations. The reconstructions can only be as good as the bottleneck representations or more likely, worse. Reconstruction is usually chosen because it is easier to interpret for biologists, because the goal of the studies is to identify features that can be used as biomarkers. If the end goal is not the classification, but to identify biomarkers that can be used to search for new therapeutical compounds, then it makes sense to focus on denoised representations. We argue, however, that the bottleneck should be used in combination with other methods, like SHAP[25] or LIME[35] (Supplementary Fig. 2), that can identify the most useful features for classification.

We found that the original NormAE available online had poor classification results on the Alzheimer dataset. Thus, instead of using the implemented version available online, it was implemented in our suite of models, as it is highly similar to the DANN method, except without the GRL (see Methods for more details), which we believe to be a crucial component to the success of the methods that use DANN. Because they are all trained with the same training scripts, we can make fair comparisons between the methods. Indeed, the original NormAE had more layers and a different training strategy than we used. All methods should be trained using the same architectures so that the only difference between each method is the method itself, not the architecture. We were able to surpass the classification performance of the default configuration that was developed for the adenocarcinoma

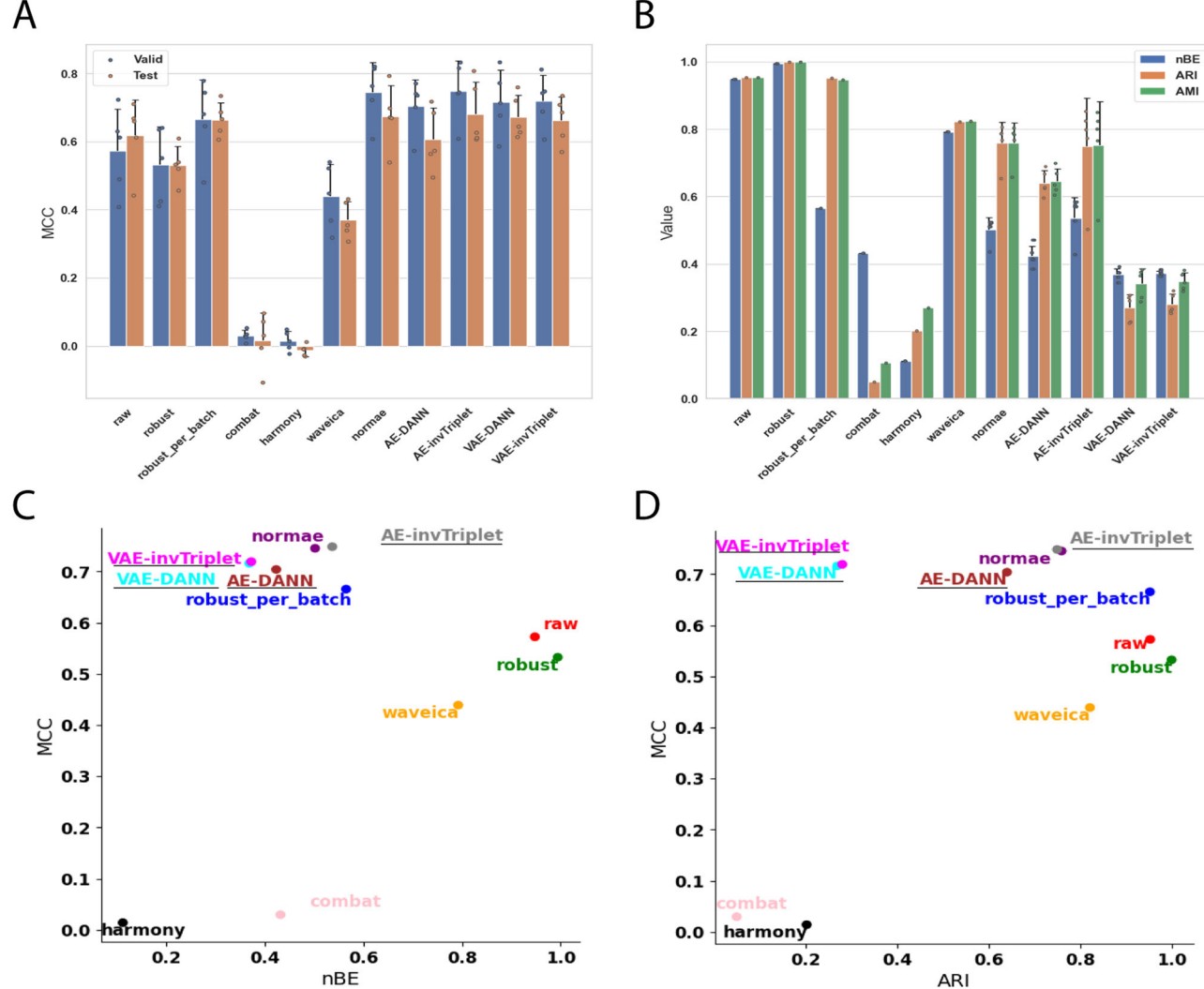

**Fig. 6 | Metrics on the Benchmark dataset. A** Valid and test MCC scores for all methods benchmarked. Higher is better. **B** Batch mixing metrics: normalized Batch Entropy (nBE), Adjusted Rand Index (ARI), Adjusted Mutual Information (AMI). Smaller is better. MCC is compared to **C** nBE and **D** ARI. Error bars represent standard deviations around the means. All error bars are derived from the results of 5-fold cross-validation ($n = 5$). The BERNN models are underlined. Source data are provided as a Source Data file.

dataset (we obtained the dataset from ref. 36, as ref. 17 did not have its dataset available, but the dataset description suggests they are the same dataset).

We made the network architecture flexible by optimizing the number of neurons per layer. They are treated as hyperparameters to be optimized, making some models trained much smaller than others, though the models are restricted to a single hidden layer for the encoder and decoder. We applied BERNN on two datasets with more than 1000 injections (the mixed tissues and benchmark datasets, with 1553 and 1027 injections, respectively). LC-MS experiments are not typically much larger than our study, so it should be able to be used without modifications for most datasets.

Other than by controlling the size of the models, we reduced overfitting by applying several regularization methods, namely weight decay, dropout[37], and label smoothing. Repeated holdout is used to make multiple train/valid combinations that are from non-overlapping batches to make the models generalizable in new batches. In experiments involving multiple batches, the data should always be split into non-overlapping batches (i.e. the training, validation, and test sets should always contain samples from the different batches), otherwise, a model may overfit for the batches seen during training and seem like it generalizes well. The model might still overfit if the different batches

used are not diverse enough. For example, if a model is generalized well using multiple batches from one research center, it may not generalize as well using batches from another research center.

In future developments, the reasons why some models are better suited for a particular dataset will be studied. For example, the VAE-based models performed much better than our other models in the dataset that had the least batch effect and the most batches, whereas it was the opposite in the dataset with the most batch effect and least number of batches. This aspect will be further developed to get improved guidelines that could be used to reduce the amount of time needed to explore the architectures and have an idea from the start of what should work in a particular framework. Additionally, we correct only for batches, but other confounding factors could be corrected. The existence of unknown confounding factors might explain why the best models for classification are not those that best correct batch effects. The methods involving DANN or Inverse Triplet Loss, which are used in BERNN to counter batch effect, could also be used to counter additional confounding effects.

In conclusion, we propose a tool, BERNN, that integrates multiple solutions to remove batch effect in LC-MS analyses while allowing an optimal classification of the biological samples in binary and multiclass problems. Using five different proteomic and metabolomic datasets,

we benchmarked BERNN models to six other tools available in the literature and found that they outperformed them in all cases while not only considering reduction of batch effect but also classification performances. Finally, at the difference of most batch correction tools which provide a corrected version of the data, here we rely on the encoded version for data classification. However, we demonstrated that combining approaches such as SHAP with BERNN can be used to retrieve key features enabling the discovery of potential biomarkers.

## Methods

### Datasets description

We are using three datasets with various levels of batch mixing heterogeneity and different numbers of batches to demonstrate how BERNN can apply to different scenarios. A summary of the three datasets is available in Fig. 2A. All preprocessed datasets are available at https://github.com/spell00/BERNN_MSMS/tree/main/data.

**Alzheimer Disease dataset.** Cerebrospinal fluid (CSF) samples were obtained according to standardized collection and processing protocols through the Mass General Institute for Neurodegenerative Disease (MIND) biorepository, following written informed consent for research biobanking (IRB: 2015P000221). This cohort represents a clinically complex cohort spanning a wide variety of neurological disorders, which closely aligns with a real-life diagnostic situation. The raw data for this dataset is publicly available on ProteomeXchange with accession number PXD043216.

CSF proteins were trypsin-digested prior to LC-MS/MS analysis on an Orbitrap Fusion Tribrid (Thermo Fischer Scientific) mass spectrometer operating in Data Independent Acquisition (DIA) mode. The 923 samples were injected in duplicate across 22 different batches. A QC sample was also generated by mixing a small aliquot of all CSF samples and analyzed in the same conditions at the beginning and at the end of each batch. This QC sample was also used to generate a Gas Phase Fractionation (GPF) library. The raw files were then processed with DIA-NN[38] software (version 1.8.1) for protein identification and quantification. DIA-NN was used in two steps: 1) Library-free search on the GPF files using a Uniprot Reference Homo Sapiens database to generate a spectral library; 2) Library-based search on the sample and QC sample files using the spectral library generated in step one. The main report generated by DIA-NN was used with the DIA-NN R package[39] to get quantifications of proteins corresponding to unique genes. Common contaminants were removed from the dataset to avoid any bias in the classification or in any subsequent analysis. A complete list of removed protein IDs is available in the supplementary content (Supplementary information file" List of contaminants").

The dataset consists of a total of 923 samples, with 84 QC samples and 839 samples obtained from 408 subjects, with 22 subjects having more than one sample from repeat clinic visits. The samples were distributed into 22 batches with an average of approximately 41 samples per batch (mean = 41.25, std = 15.18). The cohort was subdivided into 6 different disease classes. Cognitively unimpaired patients (CU), Alzheimer's Disease with dementia (DEM-AD), Alzheimer's Disease with mild cognitive impairment (MCI-AD), dementia with causes other than Alzheimer's disease (DEM-other), mild cognitive impairment with causes other than Alzheimer's disease (MCI-other) and patients with Normal Pressure Hydrocephalus (NPH). Importantly, CU patients are not healthy controls, but all had clinical indications for lumbar puncture, and span a variety of other non-dementia diagnoses. The batches are very heterogeneous with a similar total number of samples per batch, but different class compositions across each batch (Supplementary Fig. 9), and classes are not fully balanced.

**Adenocarcinoma dataset.** The dataset is composed of a total of 642 samples, with 74 QC samples and 568 patients, comprising 497 colorectal cancer and 71 chronic enteritis patients. There were 192, 192,

and 184 subject samples with 25, 25, and 24 QCs in three batches, respectively. The raw MS files were converted to mzXML using ProteoWizard[40], and then preprocessed using the R package XCMS. After data processing, the final dataset has 6461 metabolite peaks. More details on this dataset, including more thorough details on the preprocessing steps, in the original paper[36]. The preprocessed dataset is available at https://github.com/dengkuistat/WaveICA_2.0/tree/master/data.

**Aging Mice dataset.** This dataset was introduced by ref. 41. The dataset is made of 372 mice liver Multiomic profiling of the liver across diets and age in a diverse mouse population samples, of which 171 received a high-fat diet and 201 had a chow diet. There were only 3 QC samples, all in the same batch, and thus these samples were discarded. The samples were distributed into 7 batches with an average of 53 samples per batch (mean = 53.14, std = 26.91). Each sample has 17,887 features that represent the peptides' precursors. The raw data for the AgingMice dataset is available with ProteomeXchange accession number PXD009160. More information is available in the original paper that introduced the dataset[41].

The preprocessing scripts necessary to reconstruct the data matrix used in this study are found in the Github repository https://github.com/symbioticMe/batch_effects_workflow_code, which was developed to reproduce the case studies used in ref. 10, including the AgingMice dataset. The first step to reproduce the preprocessing of this dataset is to download the repository **batch_effects_workflow_code**. Then, download the file http://ftp.pride.ebi.ac.uk/pride/data/archive/2021/11/PXD009160/E1801171630_feature_alignment_requant.tsv.gz and place it in the folder *AgingMice_study/data_AgingMice/1_original_data* of the **batch_effects_workflow_code** repository. Then, run the scripts *1a_prepare_sample_annotation.R*, *1b_prepare_raw_proteome.R*, and *4b_peptide_correlation_raw_data.R* to get the matrix of log precursor values used in this study.

**Benchmark dataset.** This dataset was introduced by ref. 31 because of the difficulty of finding multi-batch datasets suitable for comparing normalization methods. We used the preprocessed data available in XLSX format at https://doi.org/10.3929/ethz-b-000545373. The data for each of the seven batches is available as separate XLSX files. We used the tables in the tab *intensities* of each XLSX file. The final dataset has a total of 1027 reads representing 72 samples, each with a varying number of replicates across all batches. Each sample has replicates in at least 2 batches and up to 7 batches (Supplementary Fig 10A). It has 6 classes: (1) AA: an amino acid mix with glycine, proline, asparagine, lysine, phenylalanine, and tyrosine. (2) FA: a fatty acid mixed with lauric acid, palmitic acid, and stearic acid. (3) A mix of nucleobases with cytosine, uracil, thymine, adenine, and guanine. (4) A mix of the three mixes. (5) A polar metabolic extract of fully 13C-labeled *Escherichia coli*, obtained by growing *E. coli* on a [U-13C]glucose minimal medium. (6) a methanol extract of the NIST SRM1950 standard reference material "metabolites in serum", was obtained by adding two volumes of methanol to one volume of reference material and centrifugation to remove the insoluble precipitate. The number of samples per batch is slightly imbalanced: the smallest batch has 135 samples and the largest batch has 156 samples. The number of samples per class is also slightly imbalanced: the class with the least samples is PP with 159 samples and the class with the most samples is Bio with 192 samples (Supplementary Fig. 10C).

**Mixed tissues dataset.** This dataset was introduced by ref. 32. 1560 data-independent acquisition (DIA)-MS runs of eight samples containing known proportions of ovarian cancer tissue, prostate cancer tissue, and yeast (strain *BY4741*) or control *HEK293T*. The data were acquired using six different Quadrupole Time-Of-Flight (QTOF) mass spectrometers operating in a single laboratory over a four-month

period. On each day of acquisition, 20 replicates were run on each instrument. This was done 13 times at spaced intervals over a four-month period. For the full protocol, please refer to ref. 32. This dataset contains two sources of unwanted variation (13 different days of acquisition and 6 different mass spectrometers used). To correct batch effects variations, each batch was defined as the samples that were run on a given machine on a day. Thus, using this definition of batches, there were a total of 78 batches. Thus, using this definition of batches, there were 78 batches. This dataset is available on ProteomeXchange with accession number PXD015912.

## Tools for evaluating batch effects

The tools we used to evaluate the presence of batch effect can be divided into 3 main categories: visual diagnostic using a dimensionality reduction technique, batch mixing metrics, and quality control metrics. It is important to note that batch effects can be subtle and difficult to detect and that different methods may identify different sources of variation in the data. Therefore, it is often recommended to use multiple methods and to carefully validate and interpret the results.

**Visual diagnostic.** The first category for evaluating batch effect is visual diagnostic (Fig. 2). This is usually done with methods such as PCA, UMAP[42], or t-SNE[43]. It is often how the batch effect is first noticed. The presence of a visually observable batch effect means it is a high source of variance in the data. However, these visualizations can be incomplete, thus the absence of a visually observable batch effect does not mean it is inexistent. For example, for the Alzheimer dataset, the batch effect is not as obvious as in other datasets. If only visual diagnostics are done, batch effects might go unnoticed.

**Batch mixing metrics.** All batch mixing tests use a classifier trained to predict which batch each sample is from. We used a k-nearest neighbors' classifier with 20 neighbors to calculate the probability of a sample belonging to each batch. The highest probabilities were used as predictions to calculate ARI and AMI. The probabilities are used to calculate the batch entropy.

**Batch Entropy.** We can say there is no batch effect when it is impossible to accurately predict from which batch a sample is drawn from. If the best prediction is random, there is no batch effect. To get the maximum batch entropy (BE), the batch classifier should predict all possible batches as equivalently probable. For example, if there are 4 batches, the batch effect is at its lowest when the highest entropy is reached, which is when the batch classifier returns the vector [0.25, 0.25, 0.25, 0.25]. To calculate batch entropy, the probability of a given sample belonging to each of the possible batches is obtained using the relative frequency of its N-nearest neighbors. The BE is given by Shannon's entropy:

$$BE = I(\boldsymbol{B}) = \sum_{i=1}^{|\boldsymbol{B}|} \log\left(\frac{1}{p(\boldsymbol{B}_i)}\right) \quad (1)$$

where $\boldsymbol{B}$ is the probability vector given by the batch classifier for a single sample.

For BE, higher values mean better batch mixing. For the metric to be easily comparable to the other two batch mixing metrics, for which decreasing values indicate better batch mixing, we made a metric we called normalized Batch Entropy (nBE), which is the maximum entropy (ME) value possible minus the BE, divided by ME. The maximum and minimum values of nBE are 1 and 0, respectively. In an experiment with K batches, the entropy is at maximum when p(B) = 1/K, thus nBE is

defined as:

$$nBE = \frac{\log(K) - BE}{\log(K)} \quad (2)$$

**Adjusted Rand index (ARI).** The Rand Index is simply the number of samples correctly identified divided by the total number of samples. It measures the similarity between two data clusters. Values close to 1 indicate high batch clustering (a KNN classifier perfectly predicts the batch each sample belongs to), so high batch mixing is represented by values close to 0 (the batch predictions of a KNN classifier are no better than a random prediction). We use the Adjusted Rand Index (ARI), which is adjusted for chance. The variables compared are the batch predictions and the batches' true values. It is defined as follows:

$$ARI = \frac{\sum_{ij}\binom{\boldsymbol{n}_{ij}}{2} - \left[\sum_i\binom{\boldsymbol{a}_i}{2}\sum_j\binom{\boldsymbol{b}_j}{2}\right]/\binom{n}{2}}{\frac{1}{2}\left[\sum_i\binom{\boldsymbol{a}_i}{2} + \sum_j\binom{\boldsymbol{b}_j}{2}\right] - \left[\sum_i\binom{\boldsymbol{a}_i}{2}\sum_j\binom{\boldsymbol{b}_j}{2}\right]/\binom{n}{2}} \quad (3)$$

where $n_{ij}$, $a_i$, $b_j$ are values from the contingency table.

**Adjusted Mutual Information (AMI).** Mutual Information measures the entropy shared between individual entropies. It measures the dependence between two variables, in this case, two discrete variables. As for ARI, values close to 1 indicate high batch clustering (a KNN classifier perfectly predicts the batch each sample belongs to), so high batch mixing is represented by values close to 0 (the batch predictions of a KNN classifier are no better than a random prediction). The variables compared are the batch predictions and the batches' true values.

$$E[MI(U,V)] = \sum_{i=1}^{R}\sum_{j=1}^{C}\sum_{\boldsymbol{n}_{ij}=(\boldsymbol{a}_i+\boldsymbol{b}_j-N)}^{\min(\boldsymbol{a}_i,\boldsymbol{b}_j)} \frac{\boldsymbol{n}_{ij}}{N}\log\left(\frac{N \cdot \boldsymbol{n}_{ij}}{\boldsymbol{a}_i\boldsymbol{b}_j}\right)$$
$$\times \frac{\boldsymbol{a}_i!\boldsymbol{b}_j!(N-\boldsymbol{a}_i)!(N-\boldsymbol{b}_j)!}{N!\boldsymbol{n}_{ij}!(\boldsymbol{a}_i-\boldsymbol{n}_{ij})!(\boldsymbol{b}_j-\boldsymbol{n}_{ij})!(N-\boldsymbol{a}_i-\boldsymbol{b}_j+\boldsymbol{n}_{ij})!}; \quad (4)$$

where $\boldsymbol{n}_{ij}$, $\boldsymbol{a}_i$, $\boldsymbol{b}_j$ are values from the contingency table. C and R are the two sets of clusters getting compared, both with N elements.

**Quality control metrics.** Two of the datasets used in this study contain QC samples that were systematically analyzed with each batch of analyses. The features of that sample should always be the same, so we can calculate how much they diverge and use these metrics to measure the batch effect importance. These metrics were (to our knowledge) introduced by[17].

**Average Pearson Correlation Coefficient (aPCC).** Because it is always the same sample, we know that all QCs should theoretically be perfectly correlated, which means a perfect Pearson Correlation Coefficient (PCC) of 1. All samples are compared in pairs, so the final value is an average overall PCCs.

**Normalized Median Euclidean Distance (nMED).** If batch correction is efficient, all QC samples should be very close to each other. The Euclidean distance is used to measure how far each pair of samples are from one another. Instead of using the average, like in ref. 17, the median is used because it is less affected by aberrant values. Unlike ref. 17, we also normalize the value by dividing it by the median Euclidean distance of all non-QC samples. If a transformation makes the QC samples very close to each other, but non-QC samples are equally close to each other, then the transformation did not actually alleviate the batch effect. For this reason, nMED should be preferred to the average Euclidean distance proposed in ref. 25.

## Batch Effect Removal methods

Apart from BERNN, we tried normalizing each dataset using three different methods: minmax, standard and robust standardization. The first methods to counter batch effects that we tried was applying the same three normalization methods, but to counter batch effects, they were applied individually to each batch individually, which we named *minmax_per_batch*, *standard_per_batch* and *robust_per_batch*. The latter two have more potential to remove the batch effect, transforming the values into z-scores, thus forcing each batch to have a mean of 0 and unit variance. We also used combat[8] and Harmony[13], as they are popular methods in microarrays/RNAseq and scRNAseq respectively. Combat is also used to remove batch effects from LC-MS datasets[10]. We used two implementations of combat: an R version https://rdrr.io/bioc/sva/man/ComBat.html and a python version https://github.com/epigenelabs/pyComBat, which we named pycombat in this manuscript. Intriguingly, the two implementations had very different outcomes.

We also used WaveICA[24] and NormAE[17], as they are state-of-the-art methods is batch effect correction of LC-MS data. We used our own implementation of NormAE, which has a slightly different architecture. We reduced the number of layers to a single hidden before and after the bottleneck to make the comparison with our own models. Unlike NormAE, we consider the number of neurons in each layer to be hyperparameters that are optimized. To give it a fair chance to outperform our own methods, we also use the same hyperparameter optimization as our models.

## Preprocessing

First, all data was logged using numpy's *log1p* function. This function is preferred to the *log* function because log1p(1) is 0, so all the missing values that we put to 0 are kept to 0 and all the non-zero values are kept positive. Then, for each of BERNN's models, we used one of the aforementioned normalization methods. Instead of choosing only just one method, we made the choice of normalization one of the hyperparameters to optimize. The choice of normalization was *standard*, *standard_per_batch*, *robust* or *robust_per_batch*.

## Autoencoder

All the models in BERNN are implemented using PyTorch and are based on autoencoders. In short, autoencoders are composed of an encoder and a decoder. The encoder turns the inputs into embeddings (also referred to as the bottleneck of the autoencoder), which are usually smaller than the inputs, but not necessarily. The objective of the autoencoder is to obtain new representations and reconstruct the original inputs from it the best it can. The embeddings should then contain as much information as possible from the inputs, without the unnecessary noise. To find the best-performing model on a given dataset, 10 models can be trained using BERNN. For a complete representation of all the possible models that can be trained using BERNN, see Supplementary Fig. 1. To reduce potential overfitting, all the autoencoders have a single hidden layer (same as in Fig. 1 and Supplementary Fig. 1). All the models were not represented in the main results to alleviate the reading, but the complete results are available in Figures S4–S6.

## Reconstruction with batch mapping

The first method that is implemented to obtain batch-free representations is to add to the embedding of the autoencoder a vector of the same size representing the batch ID (Supplementary Fig. 11). This was implemented in NormAE, although not mentioned in the original publication[17]. It makes it possible to obtain better reconstruction loss by adding the batch information into the vector for the reconstruction. Because the batch ID is contained in this vector added to the embedding, the latter does not need to contain information about the batch. A similar method is used to get batch-free representations in scRNAseq, such as in ref. 14. In this

case, the batch ID is directly appended to the bottleneck representation of the variational autoencoder.

## Domain Adversarial Neural Network (DANN).

Domain Adaptation Neural Network (DANN) is a type of deep learning algorithm that enables a model trained on one domain to be adapted to another related domain with different characteristics, allowing it to perform better on the target domain. DANN achieves this by learning to extract domain-invariant features from the input data[27]. In this case, we are defining batches to be from different domains. The original DANN was developed to adapt the learning from a single domain to another one, so our work is more akin to ref. 44, which extends domain adaptation for multiple domains. Our AE-DANN model is represented in Supplementary Fig. 1. The loss of AE-DANN is the following:

$$\min_{D,E} \min_{F_b} V(D,E,F_b) = loss_{rec}\left(x, D\left(E(x), y^b\right)\right) + \lambda^b loss_{disc_b}\left(F_b(E(x)), y^b\right)$$

(5)

The loss$_{disc-b}$ is minimized, but it is adversarial because of the Gradient Reversal Layer (GRL).

NormAE is also similar in nature to a DANN, except it does not use the GRL. Using the GRL is advantageous, because the total loss is composed of losses that are all minimized and added together. When not using the GRL, like with NormAE, the adversarial loss is maximized, and it is subtracted from the other losses being optimized simultaneously.

$$\min_{D,E} \max_{F_b} V(D,E.F_b) = loss_{rec}\left(x, D\left(E(x), y^b\right)\right) - \lambda^b loss_{disc_b}\left(F_b(E(x)), y^b\right)$$

(6)

Using this definition, if the second term of the equation becomes too large, the loss could become negative, which should not be allowed to happen.

## Domain inverse triplet loss.

The inverse triplet loss is like the normal Triplet Loss (defined in the section Reverse Triplet Loss of the supplementary material), but the positive and negative samples are inversed; the negative samples take the place of the positive samples in the triplet loss equation, and vice-versa.

$$\mathcal{L}_{invTriplet}(\mathbf{A},\mathbf{P},\mathbf{N}) = \max(|f(\mathbf{A}) - f(\mathbf{N})|_2 - |f(\mathbf{A}) - f(\mathbf{P})|_2 + \alpha, 0) \quad (7)$$

**A** is the anchor input, **P** is any Positive input of the same batch as **A, N** is any negative sample of a different batch than **A,** $\alpha$ is the margin between positive and negative pairs and $f$ is the embedding given by passing the inputs through the encoder of the autoencoder. Using the normal triplet loss would result in samples from the same batch clustering together and different batches being far away from each other. The distance between the clusters is controlled by the hyperparameter $\alpha$. The Inverse Triplet loss does the opposite by inversing the Positive and Negative samples in the equation, which encourages batch-free representations. The samples from different batches get closer, while samples from the same batch are pushed further apart. The latter objective is used to prevent all samples from collapsing. If the samples from the same batch are not pushed apart, the loss would be optimal if all samples were transformed into the exact same value, which is not the desired outcome. The distance minimized in this case is the Euclidean distance, but any distance could be used.

## Variational Autoencoders.

The variational autoencoder (VAE) is a probabilistic generative model based on the variational Bayes approach. To train a VAE, we want to optimize the lower bound defined

in Eq. 3 of ref. 45:

$$\mathcal{L}(\theta,\phi;\boldsymbol{x}^{(i)}) = -D_{KL}\left(q_\phi(\boldsymbol{z}|\boldsymbol{x}^{(i)})|p_\theta(\boldsymbol{z})\right) + E_{q_\phi(\boldsymbol{z}|\boldsymbol{x}^{(i)})}\left[\log p_\theta(\boldsymbol{x}^{(i)}|\boldsymbol{z})\right] \quad (8)$$

where $D_{KL}$ is the Kullback-Liebler Divergence, φ represents the variational parameters (encoder parameters) and θ the generative parameters (decoder parameters). The Kullback-Liebler Divergence pushes the variational posterior $q_\phi(\mathbf{z},|,\mathbf{x})$ to resemble the prior $p_\theta(\mathbf{z})$, which is the unit normal distribution. Both the labels and batch classifiers use the reparametrized variable $z$ as inputs. As depicted in Fig. 2A and Supplementary Fig. 1, z is obtained using the reparameterization trick[45] to train the layers **μ** and **σ**, which represent the average and standard deviation in the Normal distribution $z \sim p(z|x) = N(\boldsymbol{\mu},\boldsymbol{\sigma}^2)$. Because backpropagation cannot flow through random variables, it is reparameterized as $z = \boldsymbol{\mu} + \boldsymbol{\sigma}\epsilon$, where $\epsilon$ is a noise variable $\boldsymbol{\epsilon} \sim N(0,\mathbf{I})$ and $\mathbf{I}$ is the identity matrix. It is also optionally combined with a DANN, which is also trained on z to make sure the new representations are free of batch effects. All the different AEs listed above have also been implemented as VAEs, including NormAE, which is then called NormVAE.

**Total loss.** The total loss for the model is a composition of all the losses listed in this section. It can be formulated as:

$$\mathcal{L}_{total} = \mathcal{L}_{rec} + \nu \cdot \mathcal{L}_{classif} + \beta \cdot KLD + \gamma \cdot \mathcal{L}_{BE} \quad (9)$$

where $\mathcal{L}_{rec}$ is the reconstruction loss, $\mathcal{L}_{classif}$ is the labels classification loss, KLD is the Kullback-Liebler divergence and $\mathcal{L}_{BE}$ is the batch effect loss. The hyperparameters $\nu$, $\beta$, and $\gamma$ control the importance rate given to $\mathcal{L}_{classif}$, KLD and $\mathcal{L}_{BE}$, respectively. KLD only applied to the VAE models and $\mathcal{L}_{BE}$ only applies to models with a DANN, inverse Triplet Loss, and reverse Triplet Loss. Models with a DANN and reverse Triplet Loss have a $\mathcal{L}_{BE}$ that is minimized, but because of the gradient loss reversal, they increase until reaching an equilibrium at the loss that corresponds to random guessing the correct batch.

**Training strategies**

**Repeated holdout.** Repeated holdout is a method to evaluate the performance of a model on a dataset. It is similar to cross-validation, however, each split is random, there is no limit to the number of times it can be done on a dataset and the test set is resampled for every holdout iteration. Resampling the test set is particularly important because some batches can be much easier to classify than others, which can make the test set classification much better or much worse than the validation set. Using repeated holdout makes classification on the valid and test sets comparable. We use Scikit-learn's *Stratified-GroupKFold* class to make balanced splits that preserve the percentage of samples per class as much as possible, while also respecting the constraint of each split containing non-overlapping batches. This is done to detect if models generalize in new batches not seen during training.

When splitting the dataset, the samples from a given batch must all be contained in the same split. We do this to inform on the generalization abilities of a model to make predictions in a new batch. We randomly resampled the dataset five times for each dataset, except for the adenocarcinoma dataset which was resampled three times because there are only three batches (each batch was used once for the train, valid, and test splits).

**Class imbalance.** Class imbalance has a negative impact on machine learning models predictive abilities. If nothing is done about it, the model might learn to only predict the majority class. This is especially true if the imbalance is very large. It is also a concern if the problem is very hard to model. PyTorch's *WeightedRandomSampler* is used to counter class imbalances in datasets by giving more weights to samples from minority classes during training.

**BERNN Hyperparameter Optimization.** We used the function optimize from the package ax-platform (https://pypi.org/project/ax-platform/) to perform a Bayesian optimization of the hyperparameters for each of the BERNN models (all implemented in PyTorch). We used 20 combinations of hyperparameters to optimize each model. The hyperparameters optimized are the following: *learning rate, weight decay's rate, dropout rate, number of warmup epochs, layer1, layer2, label smoothing, triplet loss margin* (when it applies), *beta* (controls the strength of KL divergence, when it applies), *gamma* (controls the strength of the adversarial or triplet loss, when it applies) and which normalization to use for preprocessing (*minmax, standardization, robust standardization, minmax_per_batch, standard_per_batch, robust_per_batch*). The batch size is set to 32 and the number of epochs after warmup is set to 1000, but the training is stopped if no improvements are made in the last 100 epochs. Models were trained on Nvidia RTX3090 GPUs. A summary of all hyperparameters is found in Supplementary Table 6 and Supplementary Fig. 12 shows an example of a hyperparameter optimization of 20 iterations.

**Classification with non-BERNN representations.** In this study, we employed a diverse set of non-deep learning classifiers, namely Random Forest Classifier (RFC) and Support Vector Machine Classifier with a linear kernel (LinearSVC) to complement our suite of models based on the BERNN, which uses deep learning models. They are used either to classify the data not corrected for batch effects (*e.g.* raw or standardized data) or to classify data corrected for batch effects, either using combat[8], harmony[13], or waveica[46]. The inclusion of these non-deep learning models allowed us to establish a baseline for performance assessment. By comparing the performance of our BERNN-based models to these algorithms, we could discern whether the incorporation of deep learning techniques led to significant improvements in predictive accuracy.

For both models, we used implementations from the Python package scikit-learn (https://scikit-learn.org). To deal with classes imbalance in some datasets, the parameter *class_weights* was set to "balanced" for both models. Automatically adjust weights inversely proportional to class frequencies. The hyperparameters optimized for the RFC were *min_samples_split, min_samples_leaf, n_estimators, criterion*, and *oob_score*. For the LinearSVM, we optimized *tol, max_iter, penalty*, and *C*. The hyperparameters for the RFC and LinearSVM models were optimized using the package *scikit-optimize* (https://scikit-optimize.github.io/stable/) to execute a Bayesian Optimization. For a detailed description of all the hyperparameters used, please refer to scikit-learn's API (RFC is ensemble.RandomForestClassifier and LinearSVM is sklearn.svm.LinearSVC).

**Model interpretability.** For model explanation purposes and to identify the most important features for the classification, we used SHAP[25] to produce an example analysis. The advantage of that approach is that it makes it possible to explain the decision made on individual samples and could be used for precision medicine. It would allow the identification of complex patterns that apply only to a subset of the samples that could not be identified by differential analysis.

**Missing values**
Missing values are a common problem in LC-MS experiments. We handled them by putting them to 0. The bottleneck representation, however, won't have missing values. If the missing values are causing batch effects, it is handled by batch effect removal. Though we suggest tools such as SHAP should be more appropriate if looking for biomarkers, the reconstruction could be used for any other downstream

analysis (e.g. differential analysis) and would be as free of batch effect as the bottleneck representation.

## Reporting summary

Further information on research design is available in the Nature Portfolio Reporting Summary linked to this article.

## Data availability

All the preprocessed data necessary to reproduce the experiments can be found in the repository https://github.com/spell00/BERNN_MSMS/tree/nature_comm_2023/data in the folder named 'data'. Source data are provided in this paper. The raw data for the Alzheimer dataset is available on ProteomeXchange with accession number PXD043216. Source data are provided in this paper.

## Code availability

Notebooks and Python scripts used for data visualization, batch effect correction, and classification are available at https://github.com/spell00/BERNN_MSMS[47].

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

## Acknowledgements

The AD research laboratory is supported by Research and Innovation Chair L'Oréal in Digital Biology. The Orbitrap Fusion mass spectrometer utilized was supported in part by NIH SIG grants S10OD018034 and Yale School of Medicine. Funding for this project came from NIH awards AG062421 to SEA, AG062306 to SEA, BCC & ACN, and AG066508 to ACN. BCC is supported by an Alzheimer's Research UK Senior Research Fellowship, and by the Bright Focus Foundation.

## Author contributions

S.P. conceived the study, conducted literature analysis, conceived the models, designed the methodology, wrote all the code, conducted all the experiments, analyzed the results, and wrote the draft manuscript. S.P., F.P., and M.L. contributed to the design of the methodology. S.P., M.L., F.R.D., F.P., B.C., M.B.G., A.D. reviewed, commented, and revised the final manuscript. S.L., B.C., M.B.G., W.W., and T.T.L. generated the LC-MS/MS data for the Alzheimer Dataset. F.R.D. processed the raw data for the Alzheimer dataset. A.C.N., S.E.A., and B.C. acquired the funding, designed the study, and supervised the acquisition of the Alzheimer dataset. A.D., M.L., F.R.D., and F.P. supervised the study.

## Competing interests

S. Arnold has received honoraria and/or travel expenses for lectures from Abbvie, Eisai, and Biogen and has served on scientific advisory boards of Corte, has received consulting fees from Athira, Cassava, Cognito Therapeutics, EIP Pharma and Orthogonal Neuroscience, and has received research grant support from NIH, Alzheimer's Association, Alzheimer's Drug Discovery Foundation, Abbvie, Amylyx, EIP Pharma, Merck, Janssen/Johnson & Johnson, Novartis, and vTv. S.N. Leslie is a current employee of Janssen Pharmaceuticals. B. Carlyle has received grant funding from Ono Pharmaceutical. Other authors report no conflicts of interest.
