## [Peer Review File · Nature Communications]

Reviewers' Comments:

Reviewer #1:

Remarks to the Author:

The authors developed a suite of batch correction methods and termed it as BERNN, which can remove batch effects from data generated by LC-MS experiments. The aim is to maximize sample classification performance between conditions.

1. In the abstract, it is mentioned that "Correcting batch effects is crucial for the reproducibility of proteomics research, but current methods..." However, this study covers more than just proteomics, right? The three datasets used comprise proteomic, peptide and metabolic data.
2. Unlike single-cell experiments, LC-MS experiments typically have smaller sample sizes. The three datasets used here contain sample numbers ranging from 372 to 642 (or 839 if duplicates are counted). The manuscript would benefit from a clearer explanation of why deep learning methods are suitable for batch correcting LC-MS experiments, especially since lines 78-81 points out a reason why deep learning might not be appropriate.
3. For large-scale DIA-MS experiments with more than a thousand samples, there are two studies: <https://www.nature.com/articles/s41467-020-17641-3> and [https://www.cell.com/cancer-cell/fulltext/S1535-6108\(22\)00274-4](https://www.cell.com/cancer-cell/fulltext/S1535-6108(22)00274-4). The former article explicitly addresses reproducibility issues in LC-MS, including batch effects. Will BERNN work on these large-scale proteomic datasets?
4. It's worth mentioning that missing values are a significant issue in LC-MS experiments, as they directly contribute to batch effects. However, this aspect was not addressed in the manuscript.
5. The manuscript states, "These methods assume an accurate new representation can be obtained using a generalized linear model, which is not necessarily an accurate assumption." It would be helpful to provide data or citations to support this claim and explain how and why DNN can address this problem.
6. The description of the three datasets appears to be unbalanced, with the first dataset (5.1.1) being the most comprehensive. It would be beneficial to provide a rationale for this discrepancy.
7. The manuscript focuses on classification performance as one of the downstream tasks. However, it is unclear how BERNN will work on other tasks such as differential expression analysis and survival analysis. Additionally, all three presented cases involve basic binary classification, and it could be interesting to explore multi-class classification problems.
8. Consistency in terminology should be maintained throughout the manuscript, e.g., choose either SHAP or Shapley and use it consistently.
9. The term "repetitive holdout" should be "repeated holdout" or "Monte-Carlo simulation", right?
10. Some of the mathematical notations in Figure 1 are not explained in the text. It would be helpful to provide the necessary explanations for all symbols used.
11. In Table A (Figure 2), the precursors of Aging Mice and metabolic data of Adeno are not proteins (second last column).

Reviewer #2:

Remarks to the Author:

The article (NCOMMS-23-28695-T) titled "Enhancing Classification of liquid chromatography mass spectrometry data with Batch Effect Removal Neural Networks (BERNN)" presents machine learning models that are shown to outperform six other previously published methods in sample classification across three datasets. It is reported that the optimal model for classification does not always excel in batch effect removal, highlighting a trade-off between these aspects. As expected, excessive batch effect correction could lead to the loss of essential biological information, and it is crucial to consider this in such approaches. The results are therefore noteworthy for the field, as LC-MS technology is known to suffer from batch effects. BERNN models can aid in identifying significant results and show promise for improving LC-MS data classification while considering dataset characteristics.

Variations in sample processing protocols, experimental conditions, and data acquisition techniques are known to prevent genuine biological signals from being detected, and the authors have chosen three diverse proteomic and metabolomic datasets to test their models. Although one might argue that these datasets are far from representing the entire scope of different LC-MS experiments, they do provide a ground for the authors to show that BERNN can outperform other

batch effect correction techniques in both batch effect attenuation and classification performance, and is a welcome addition to the tools available in the field. BERNN's resilience to varying hyperparameter configurations should be explored in further details to facilitate more informed decisions for researchers considering BERNN adoption in their own studies.

The data analysis, interpretation, and conclusions are sound. Nonetheless, minor enhancements could enhance the publication. These include more elaboration on the hyperparameters employed and further discussion on the potential for overfitting in BERNN models and how users can make sure to avoid such issues. Overall, while well-structured and supported by robust evidence, the article could benefit from these revisions, which would enhance its utility within the research community.

The employed methodology in the article adheres to established field standards and demonstrates robustness. Evaluation included a comprehensive array of metrics, encompassing batch effect removal and classification performance. To mitigate potential biases, the authors judiciously adopted a blinded approach during method assessment, ensuring evaluators remained unaware of the method in use.

The methods section of the article offers a commendable overview of the study's procedural steps, and the code repository mentioned in the article has documentation for reproducing the results. However, there exists potential for greater detail and more comprehensive information regarding each of the machine learning algorithms employed and the libraries utilized for their implementation, more detailed insight into the hyperparameters utilized during model training, further elaboration on the pre-processing procedures applied to the data, incorporating steps taken for missing value handling, data normalization, and data transformation. These enhancements should also be complemented by a more comprehensive step-by-step guide or video tutorial to facilitate less code-savvy biologists to be able to benefit from BERNN.

REVIEWER COMMENTS

Reviewer #1 (Remarks to the Author):

The authors developed a suite of batch correction methods and termed it as BERNN, which can remove batch effects from data generated by LC-MS experiments. The aim is to maximize sample classification performance between conditions.

1. In the abstract, it is mentioned that "Correcting batch effects is crucial for the reproducibility of proteomics research, but current methods..." However, this study covers more than just proteomics, right? The three datasets used comprise proteomic, peptide and metabolic data.

Answer: The study indeed encompasses more than just proteomics. In addition to proteomic data from two datasets, it also incorporates metabolic data from one dataset. The term proteomics was changed to omics in the abstract (page 2).

2. Unlike single-cell experiments, LC-MS experiments typically have smaller sample sizes. The three datasets used here contain sample numbers ranging from 372 to 642 (or 839 if duplicates are counted). The manuscript would benefit from a clearer explanation of why deep learning methods are suitable for batch correcting LC-MS experiments, especially since lines 78-81 points out a reason why deep learning might not be appropriate.

Answer: Deep learning models indeed carry a significant risk of overfitting. To mitigate this concern, we adopted a rigorous approach, which included meticulous model selection, the incorporation of regularization techniques such as weight decay, dropout, and label smoothing, fine-tuning of hyperparameters, and the implementation of a repeated holdout strategy. In this strategy, we resampled the validation and test sets up to 5 times, thereby substantially diminishing the likelihood of obtaining favorable results solely by chance.

In our experiments, we consistently observed that batch effects could be reduced using BERNN or even conventional techniques like combat. However, achieving the highest batch correction scores often involved the consequence of removing valuable biological signals essential for classification tasks. This is a key motivation for adopting neural networks over traditional methods, like combat or harmony, because they offer the flexibility to partially address batch effects while preserving the optimal classification model.

Our investigations revealed that combat or harmony batch effect removal methods failed to yield improvements in the classification task, and in some cases, worsened the

classification performance. This underscores the inherent tradeoff between batch effect mitigation and the preservation of biologically meaningful variation, a tradeoff that remains unaddressed in non-deep learning models.

Finally, through rigorous model testing and refinement, we were able to surmount the overfitting challenge and ultimately enhance classification performance, particularly in handling new batches, surpassing the capabilities of other batch correction methods.

3. For large-scale DIA-MS experiments with more than a thousand samples, there are two studies: <https://www.nature.com/articles/s41467-020-17641-3> and [https://www.cell.com/cancer-cell/fulltext/S1535-6108\(22\)00274-4](https://www.cell.com/cancer-cell/fulltext/S1535-6108(22)00274-4). The former article explicitly addresses reproducibility issues in LC-MS, including batch effects. Will BERNN work on these large-scale proteomic datasets?

Answer: We have a high level of confidence in BERNN's compatibility with these extensive proteomic datasets. It's well-established that neural networks tend to exhibit improved performance as they are provided with larger volumes of data, a principle that applies to machine learning models in general.

4. It's worth mentioning that missing values are a significant issue in LC-MS experiments, as they directly contribute to batch effects. However, this aspect was not addressed in the manuscript.

Answer: Missing values were addressed by imputing them with zeros. However, the bottleneck representation has no missing values. In cases where missing values contribute to batch effects, our batch effect removal approach effectively manages this aspect. Our method is designed to identify and extract a batch-effect-free representation, if such a representation exists, without being constrained by linear transformations. To provide additional clarity on this matter, we have incorporated a dedicated section on handling missing values in the methods section of the manuscript (section 5.7, page 18).

5. The manuscript states, "These methods assume an accurate new representation can be obtained using a generalized linear model, which is not necessarily an accurate assumption." It would be helpful to provide data or citations to support this claim and explain how and why DNN can address this problem.

Answer: This statement is supported by citation 14, which was initially cited in the following sentence but has been reiterated here for enhanced clarity. Moreover, we have introduced the notion that linear models may not comprehensively capture the intricate

nature of batch effects, particularly when they exhibit nonlinearity. It's worth noting that the Deep Neural Network addresses the limitations of using a generalized linear model by employing a sequence of nonlinear transformations, a concept also supported by citation 14 (Lopez et al, 2014). (page 4)

6. The description of the three datasets appears to be unbalanced, with the first dataset (5.1.1) being the most comprehensive. It would be beneficial to provide a rationale for this discrepancy.

Answer: The primary dataset receives a more extensive description because it marks the initial publication utilizing this specific dataset. Conversely, the other two datasets have been comprehensively detailed in their original publications, which are properly cited. Our additional information primarily serves to provide the reader with essential insights, such as the dataset's sample count, the distribution of samples across batches, the inclusion of quality control samples (if applicable), the total number of features in the dataset, references to more comprehensive dataset documentation, and guidance on how to access the dataset.

7. The manuscript focuses on classification performance as one of the downstream tasks. However, it is unclear how BERNN will work on other tasks such as differential expression analysis and survival analysis. Additionally, all three presented cases involve basic binary classification, and it could be interesting to explore multi-class classification problems.

Answer: In our text, we have explicitly recommended using Shapley values for biomarker discovery. To reinforce this recommendation, we have introduced clarifications in both the introduction's final paragraph and in section 5.5.5, which focuses on model interpretability. These additions stipulate that we do not endorse differential expression analysis as a downstream task of BERNN, nor as survival analysis. We've made efforts to convey that while it's technically feasible to employ BERNN for this purpose, we advise against using the reconstruction for this specific task. The decision to utilize the bottleneck of the autoencoder for classification instead of reconstruction was based on the bottleneck's reduced size, which helps mitigate overfitting. If one were to use differential expression for biomarker discovery, the reconstruction would indeed be necessary. However, it's crucial to recognize that the reconstruction loss is a secondary objective compared to the primary classification objective.

Our choice to exclusively focus on binary classifications was made to showcase the effectiveness of our method. Nonetheless, BERNN is applicable to multi-class problems as well. While we did not explore multi-class scenarios in this study, we anticipate that future research, whether conducted by us or others, will delve into the application of our

methods for this purpose. Demonstrating the effectiveness of BERNN in the context of multi-class problems would be indeed essential, and we have taken care to emphasize in the text that BERNN is suited for such applications (section 5.1, page 11).

8. Consistency in terminology should be maintained throughout the manuscript, e.g., choose either SHAP or Shapley and use it consistently.

Answer: To ensure terminological consistency in the manuscript, we have uniformly replaced "Shapley" with "SHAP."

9. The term "repetitive holdout" should be "repeated holdout" or "Monte-Carlo simulation", right?

Answer: You are correct, it should indeed be "repeated holdout." Consequently, we have made the necessary adjustments by replacing all instances of "repetitive holdout" with "repeated holdout" throughout the manuscript.

10. Some of the mathematical notations in Figure 1 are not explained in the text. It would be helpful to provide the necessary explanations for all symbols used.

Answer: Indeed, some of the mathematical notations in Figure 1 were not adequately explained in the text. To rectify this, we have included the necessary explanations for all symbols used in the methods section (section 5.5.4, page 17).

11. In Table A (Figure 2), the precursors of Aging Mice and metabolic data of Adeno are not proteins (second last column).

Answer: Thank you for pointing that out. To accurately represent the data, we have updated the name of the column to "Features" in Table A (Figure 2), ensuring clarity regarding the nature of the listed components.

Reviewer #2 (Remarks to the Author):

The article (NCOMMS-23-28695-T) titled "Enhancing Classification of liquid chromatography mass spectrometry data with Batch Effect Removal Neural Networks (BERNN)" presents machine learning models that are shown to outperform six other previously published methods in sample classification across three datasets. It is reported that the optimal model for classification does not always excel in batch effect removal, highlighting a trade-off between these aspects. As expected, excessive batch effect correction could lead to the loss of essential biological information, and it is crucial to consider this in such approaches. The results are therefore noteworthy for the field, as LC-MS technology is known to suffer from batch effects. BERNN models can aid in identifying significant results and show promise for improving LC-MS data classification while considering dataset characteristics.

Variations in sample processing protocols, experimental conditions, and data acquisition techniques are known to prevent genuine biological signals from being detected, and the authors have chosen three diverse proteomic and metabolomic datasets to test their models. Although one might argue that these datasets are far from representing the entire scope of different LC-MS experiments, they do provide a ground for the authors to show that BERNN can outperform other batch effect correction techniques in both batch effect attenuation and classification performance, and is a welcome addition to the tools available in the field. BERNN's resilience to varying hyperparameter configurations should be explored in further details to facilitate more informed decisions for researchers considering BERNN adoption in their own studies.

The data analysis, interpretation, and conclusions are sound. Nonetheless, minor enhancements could enhance the publication. These include more elaboration on the hyperparameters employed and further discussion on the potential for overfitting in BERNN models and how users can make sure to avoid such issues. Overall, while well-structured and supported by robust evidence, the article could benefit from these revisions, which would enhance its utility within the research community.

The employed methodology in the article adheres to established field standards and demonstrates robustness. Evaluation included a comprehensive array of metrics, encompassing batch effect removal and classification performance. To mitigate potential biases, the authors judiciously adopted a blinded approach during method assessment, ensuring evaluators remained unaware of the method in use. The methods section of the article offers a commendable overview of the study's procedural

steps, and the code repository mentioned in the article has documentation for reproducing the results. However, there exists potential for greater detail and more comprehensive information regarding each of the machine learning algorithms employed and the libraries utilized for their implementation, more detailed insight into the hyperparameters utilized during model training, further elaboration on the pre-processing procedures applied to the data, incorporating steps taken for missing value handling, data normalization, and data transformation. These enhancements should also be complemented by a more comprehensive step-by-step guide or video tutorial to facilitate less code-savvy biologists to be able to benefit from BERNN.

- 1- BERNN's resilience to varying hyperparameter configurations should be explored in further details to facilitate more informed decisions for researchers considering BERNN adoption in their own studies.**

Answer: To provide a more comprehensive understanding of BERNN's adaptability to diverse hyperparameter configurations, we have introduced a new figure (Figure S10) displaying parallel coordinates. This addition serves as an illustrative example of classification scores and aims to support researchers in making well-informed decisions regarding the adoption of BERNN in their own studies.

- 2- Minor enhancements could enhance the publication. These include more elaboration on the hyperparameters employed and further discussion on the potential for overfitting in BERNN models and how users can make sure to avoid such issues.**

Answer: Minor enhancements have been made to enrich the publication. Initially, the hyperparameters were presented with limited detail in the section "BERNN hyperparameter optimization." To address this, we have included a new table (Table S2) providing comprehensive descriptions of all hyperparameters.

Additionally, one of the reviewers raised the concern of overfitting, a challenge often encountered in Deep Learning models. In response, we have incorporated a new paragraph in the discussion (6th paragraph of the discussion, beginning with "Other than by controlling the size of the models", page 10) to address this issue and provide insights into mitigating overfitting concerns.

- 3- There exists potential for greater detail and more comprehensive information regarding:**
 - a. Each of the machine learning algorithms employed**

Answer: In section 5.4.4 (page 17), we added a more detailed description of the machine learning models used (other than BERNN) (section 5.6.4, page 19).

b. Each of the libraries utilized for their implementation

Answer: We added a table in the supplementary about the R and python packages used. (Table S3)

c. The hyperparameters utilized during model training

Answer: A new table was added (table S2) was added to give more details on the hyperparameters used.

d. Elaboration on the pre-processing procedures applied to the data, incorporating steps taken for missing value handling, data normalization, and data transformation

Answer: We added a new section to describe how missing values were handled. (section 5.7, page 19)

We added a section to describe how the data is normalized and transformed to log values (section 5.4, page 15).

4- These enhancements should also be complemented by a more comprehensive step-by-step guide or video tutorial to facilitate less code-savvy biologists to be able to benefit from BERNN.

Answer: To ensure that BERNN is accessible to a broader audience, including biologists with limited coding experience, we have taken steps to enhance the README documentation. The improved README now offers a more comprehensive guide, accompanied by multiple examples demonstrating how to run BERNN on both the datasets from this study and custom datasets. While the use of a terminal is still required, these enhancements should substantially simplify the process for less code-savvy biologists, making it a more user-friendly experience.

Reviewers' Comments:

Reviewer #1:

Remarks to the Author:

In response to question 3, the authors asserted that their method would be effective with the two datasets I suggested, but this claim was only made verbally. My question may not have been clear enough. Essentially, I expected the authors to practically demonstrate the efficacy of their method on at least one of these datasets.

Similarly, regarding question 7, the authors merely stated, without practical demonstration, that their method would be applicable to multi-class problems. Providing an example would have been more convincing.

Reviewer #2:

Remarks to the Author:

The authors propose a novel Batch Effect Removal Neural Network (BERNN) methodology to remove batch effects in large LC-MS experiments while preserving biological diversity. They demonstrate that BERNN outperforms existing methods in terms of sample classification performance. However, the model with the best classification performance was not always the best at removing batch effects, and overcorrection of batch effects can lead to a loss of biological information. These findings highlight the importance of finding a balance between batch effect removal and preserving biological diversity.

The work is likely to be of significance to the field of proteomics, as batch effects are a major challenge in this field. The BERNN methodology appears to be sound, and the results are promising. Overall, the work is well-written and the results are promising. The methodology is sound and the conclusions are supported by the data.

The authors have implemented the suggestions provided in the initial review in regards to resilience to varying hyperparameter configurations, the potential for overfitting in BERNN models, the machine learning algorithms employed and libraries utilized for their implementation, the pre-processing procedures applied to the data, steps taken for missing value handling, data normalization, and data transformation, and have provided more comprehensive details on running the code.

REVIEWER COMMENTS

Reviewer #1 (Remarks to the Author):

In response to question 3, the authors asserted that their method would be effective with the two datasets I suggested, but this claim was only made verbally. My question may not have been clear enough. Essentially, I expected the authors to practically demonstrate the efficacy of their method on at least one of these datasets.

Similarly, regarding question 7, the authors merely stated, without practical demonstration, that their method would be applicable to multi-class problems. Providing an example would have been more convincing.

Answer: We have analyzed one of the datasets suggested. We chose to include an analysis of the dataset presented in <https://www.nature.com/articles/s41467-020-17641-3> . We chose this one and not [https://www.cell.com/cancer-cell/fulltext/S1535-6108\(22\)00274-4](https://www.cell.com/cancer-cell/fulltext/S1535-6108(22)00274-4) because the authors in the later reported no batch effects. It had 1560 samples and a total of 8 classes. We added a description of the dataset in the supplementary. The main goal of our approach is to improve classification by removing batch effects. The classification on the raw data was almost perfect, which did not leave enough space to observe significant improvements. Nevertheless, many other approaches for correcting batch effect had a reduction in classification performance. All BERNN's models were able to remove a lot of batch effect while maintaining nearly perfect classification scores (Figure S7).

Reviewer #2 (Remarks to the Author):

The authors propose a novel Batch Effect Removal Neural Network (BERNN) methodology to remove batch effects in large LC-MS experiments while preserving biological diversity. They demonstrate that BERNN outperforms existing methods in terms of sample classification performance. However, the model with the best classification performance was not always the best at removing batch effects, and overcorrection of batch effects can lead to a loss of biological information. These findings highlight the importance of finding a balance between batch effect removal and preserving biological diversity. The work is likely to be of significance to the field of proteomics, as batch effects are a major challenge in this field. The BERNN methodology appears to be sound, and the results are promising. Overall, the work is well-written and the results are promising. The methodology is sound and the conclusions are supported by the data.

The authors have implemented the suggestions provided in the initial review in regards to resilience to varying hyperparameter configurations, the potential for overfitting in BERNN models, the machine learning algorithms employed and libraries utilized for their implementation, the pre-processing procedures applied to the data, steps taken for missing value handling, data normalization, and data transformation, and have provided more comprehensive details on running the code.

Answer: Reviewer #2 did not ask any modifications in this round.

Reviewer #2 (Remarks on code availability):

I have looked at the repository, and the documentation is now more elaborate and complete compared to the previous submission. However, I have not tried to run and test the software.

Reviewers' Comments:

Reviewer #1:

Remarks to the Author:

The authors have addressed all my comments.